# Differential effects of visual versus auditory biofeedback training for voluntary postural sway

**Naoya Hasegawa**[1], **Kenta Takeda**[2], **Martina Mancini**[3], **Laurie A. King**[3], **Fay B. Horak**[3], **Tadayoshi Asaka**[1] *

**1** Faculty of Health Sciences, Department of Rehabilitation Science, Hokkaido University, Sapporo, Hokkaido, Japan, **2** Department of Rehabilitation for the Movement Functions, Research Institute of National Center for Persons with Disabilities, Tokorozawa, Saitama, Japan, **3** Department of Neurology, Oregon Health & Science University, Portland, Oregon, United States of America

* ask-chu@hs.hokudai.ac.jp

**Data Availability Statement:** All relevant data are within the paper.

**Funding:** This study was supported by Grant-in-Aid for Early-Career Scientists (No. 20K19371, NH) and

## Abstract

Augmented sensory biofeedback training is often used to improve postural control. Our previous study showed that continuous auditory biofeedback was more effective than continuous visual biofeedback to improve postural sway while standing. However, it has also been reported that both discrete visual and auditory biofeedback training, presented intermittently, improves bimanual task performance more than continuous visual biofeedback training. Therefore, this study aimed to investigate the relative effectiveness of discrete visual biofeedback versus discrete auditory biofeedback to improve postural control. Twenty-two healthy young adults were randomly assigned to either a visual or auditory biofeedback group. Participants were asked to shift their center of pressure (COP) by voluntary postural sway forward and backward in line with a hidden target, which moved in a sinusoidal manner and was displayed intermittently. Participants were asked to decrease the diameter of a visual circle (visual biofeedback) or the volume of a sound (auditory biofeedback) based on the distance between the COP and the target in the training session. The feedback and the target were given only when the target reached the inflection points of the sine curves. In addition, the perceptual magnitudes of visual and auditory biofeedback were equalized using Stevens' power law. Results showed that the mean and standard deviation of the distance between COP and the target were reduced int the test session, removing the augmented sensory biofeedback, in both biofeedback training groups. However, the temporal domain of the performance improved in the test session in the auditory biofeedback training group, but not in the visual biofeedback training group. In conclusion, discrete auditory biofeedback training was more effective for the motor learning of voluntarily postural swaying compared to discrete visual biofeedback training, especially in the temporal domain.

for Scientific Research (No. 18K10702, TA) from Japan Society for the Promotion of Science (JSPS), the National Institutes of Health under award (No. R01AG006457, PI: FBH), and Department of Veterans Affairs Merit Award (No. 5I01RX001075, PI: FBH). The funders had no role in study design, data collection and analysis, decision to publish, or preparation of the manuscript.

**Competing interests:** The authors have declared that no competing interests exist.

## Introduction

Augmented sensory biofeedback has been used for decades to train an individual to use his/her own physiological behavior for the purpose of improving performance. The biofeedback systems for postural control aim to provide additional sensory information about postural equilibrium or orientation to the central nervous system [1,2]. Various forms of biofeedback, including visual and auditory, have been suggested to be beneficial for improving postural control in healthy or neurological cohorts [3–5].

Previous studies have reported that both visual and auditory biofeedback improve postural control during quiet and perturbed stance, as well as gait [6–14]. These results were obtained with continuous biofeedback, where the visual or auditory information was restituted continuously to the user, as opposed to intermittently (discrete). However, the use of continuous biofeedback, particularly visual, seems to result in excessive dependence on the augmented sensory biofeedback, as revealed by performance deterioration upon its removal [2,15,16]. In fact, Lakhani and Mansfield [11] reported that a continuous visual biofeedback, displaying the center of pressure (COP) time-series data on a monitor, successfully reduced postural sway during standing on a foam surface; however, the effects were not maintained when the augmented sensory biofeedback was removed. On the contrary, continuous auditory biofeedback, that provided changing volume and frequency of tones correlated with COP displacements and directions, reduced postural sway during quiet stance even after the augmented sensory biofeedback was removed [6–10]. Although a few studies have reported the effects of visual or auditory biofeedback training on postural control, in our knowledge, only our previous study reported that one modality was better than the other by direct comparison.

The previous study reported different learning effects resulting from continuous auditory biofeedback training compared to continuous visual biofeedback training during a voluntary postural control task in which subjects aimed to follow a moving target with their body sway [17]. Specifically, the performance, such as timing accuracy relative to the target, was superior after continuous auditory biofeedback training compared to continuous visual biofeedback training, when the augmented sensory biofeedback was removed. In addition, the training effects were retained 48 hours after the biofeedback training, suggesting a learning effect. Recently, Chiou et al. [18] compared the learning effects of continuous or discrete visual biofeedback training and discrete auditory biofeedback training during a bimanual coordination task, such as the 90˚-out-of-phase, bimanual coordination pattern. They reported that both discrete visual and auditory biofeedback training resulted in better performance compared to the continuous visual biofeedback training when the augmented sensory biofeedback was removed. However, no significant differences were found between the discrete visual and auditory biofeedback training. The researchers concluded that the different learning effects after biofeedback training was modulated not only by the modalities of biofeedback (visual biofeedback versus auditory biofeedback) but also by the type of information (continuous biofeedback versus discrete biofeedback). However, the study by Chiou et al. [18] investigated the learning effects only in the spatial domain, such as spatial accuracy relative to the target, but not the temporal domain, such as the correlation between actual movements and the ideal movement. Furthermore, it is unknown whether similar learning effects would be achieved using discrete visual biofeedback and discrete auditory biofeedback for postural control.

The goal of this study was to investigate the learning effects of discrete auditory versus visual biofeedback to improve postural control, using a voluntary postural sway task [17]. A previous study using functional magnetic resonance imaging showed that brain activation increased in sensory-specific areas during visual biofeedback training. In contrast, brain activation gradually decreased over time with auditory biofeedback training [19]. These findings

suggest that auditory biofeedback training may suppress reliance on augmented biofeedback during training unlike visual biofeedback training which requires sustained dependence on vision. Moreover, previous studies showed that auditory inputs are processed more quickly, shorter reaction times, compared to visual inputs for motor response [20–22]. Thus, auditory biofeedback would result in faster, more accurate influence on the temporal domain of postural control compared to visual biofeedback. Therefore, we hypothesized that discrete auditory biofeedback training would result in better learning effects than visual biofeedback, especially in the temporal domain for control of voluntary postural sway.

## Materials and methods

### Participants

Twenty-two healthy young adults (aged 19 to 23) with no known neurological or musculoskeletal disorders participated in this study. The participants were randomly assigned to either auditory biofeedback or visual biofeedback group. Exclusion criteria for both groups were: any neurological or musculoskeletal impairments, or any auditory or visual disabilities that would interfere with balance, or with following instructions. This study was approved by the Hokkaido University Ethics Committee (Project number 16–47). Prior to their inclusion participants were informed about the experimental protocol and gave their written informed consent. All works were conducted in accordance with the declaration of Helsinki (1964).

### Equipment

A force plate (Kistler, Model 9286A, Winterthur, Switzerland) was used to calculate the COP coordinates in the anteroposterior (AP) direction. Force plate data were collected at a sampling frequency of 1000 Hz and filtered with a fourth-order 10-Hz low-pass zero-lag Butterworth filter. Real-time biofeedback was provided on a 19-inch monitor (visual) or by two speakers (auditory) located approximately 1 m from the participant. Biofeedback was programmed using LabVIEW version 2016 (The National Instruments Corp., Austin, TX, USA).

### Procedure

Participants were instructed to stand barefoot with their arms crossed on their chest, and their feet parallel and positioned 1 cm medial to the right or left anterior superior iliac spine [23]. To measure the stability limits in the AP direction, participants were asked to stand still for 5 seconds before they were asked to lean in the forward direction as far as they could, and to hold the maximum COP position for 30 seconds using a visual point on the monitor indicating COP displacement. The same procedure was then repeated in the backward direction. We trained postural control in the AP direction to reduce feedback complexity and allow participants to focus on COP fluctuations along a single axis [13]. The point moved upward on the monitor, located at eye level, as the COP moved forward and vice versa. After measuring the stability limits, participants were asked to perform the test and training sessions with the same stance and position of arms while maintaining attention on the monitor.

### Test sessions

The participant performed 5 test sessions: before and after first training (pre-1 and post-1), before and after second training (pre-2 and post-2), and 48 hours after the second training session (retention) (Fig 1). Participants were asked to track real-time body COP displacements in line with a moving target. First, the target moved to 80% of the stability limits in the forward direction in each participant, and then, moved back to 70% of the stability limits in the

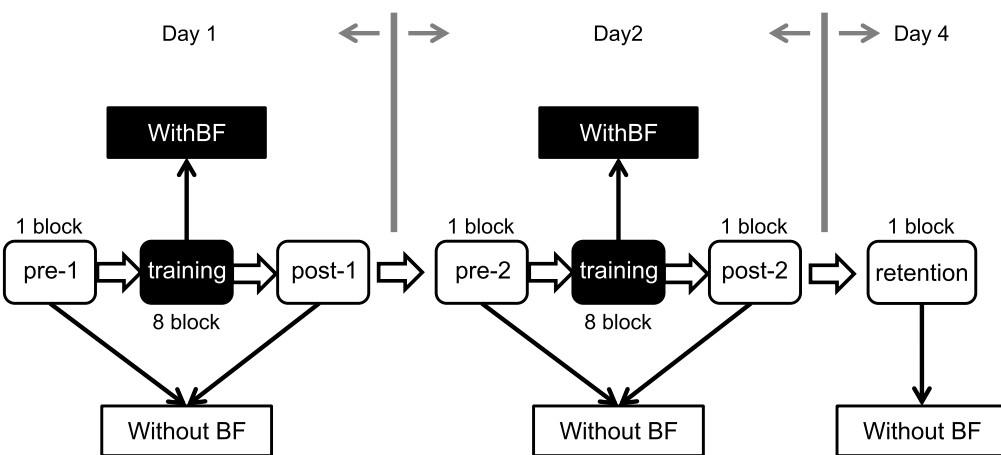

**Fig 1. Study design.** Participants were randomized into one of two groups: discrete visual biofeedback or discrete auditory biofeedback. The white boxes indicate five test sessions, and black boxes indicate two training sessions. The participants performed four test sessions: pre-1 and post-1 on the first day (Day 1), pre-2 and post-2 on the second day (Day 2), and retention on the fourth day after training (Day 4) without augmented sensory biofeedback. The training sessions consisted of 8 blocks with augmented sensory biofeedback. One block consisted of 5 trials, and each trial (seven cycles) had a duration of 35 seconds. BF, augmented sensory biofeedback.

backward direction in each participant. The movements of the target consisted of sine curves at 0.23 Hz [17,24] and repeated seven cycles for 30 seconds in each trial. A red-colored circle became visible at the center of the monitor in synchronization with a beeping sound only when the target reached the sine-wave inflection points (hidden target, see Fig 2). To calculate the start positions of the target, participants were asked to stand still for 5 seconds, and then

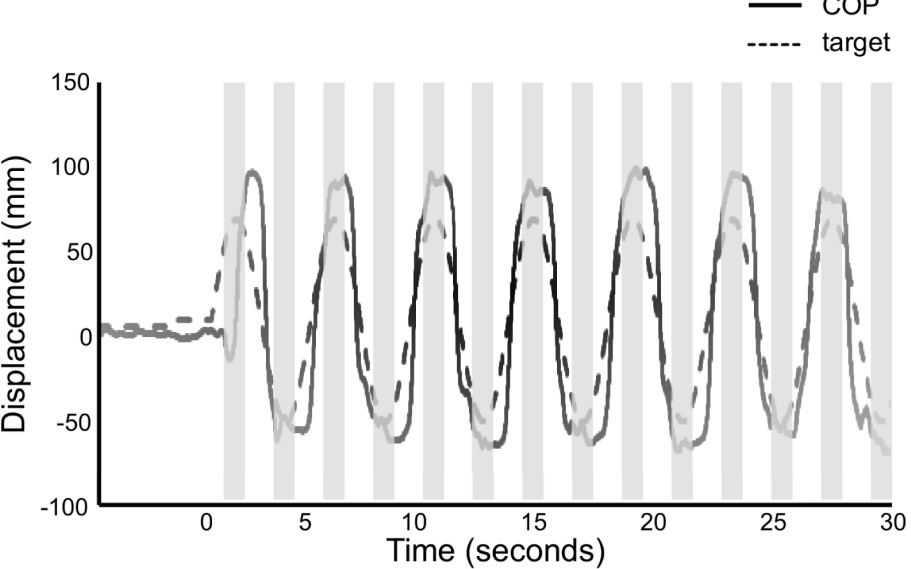

**Fig 2. Representative example of the target movements and the displacements of center of pressure.** The black solid line represents the displacements of center of pressure (COP), and the black dashed line represents the movements of target in anteroposterior directions. Participants received visual and auditory cue, and augmented sensory biofeedback. Augmented sensory biofeedback was presented 75 milliseconds before and after the moving target reached the inflection points of sine curves. The gray-colored areas represent the time intervals of biofeedback. BF, augmented sensory biofeedback.

they saw a black-colored circle on the monitor with a beeping sound as a start signal of the movements of the target. The start position of the target was averaged from the COP displacements during the first 5 seconds of each trial using the customized program of LabVIEW.

## Training sessions

The participants of both groups performed 80 trials across two consecutive days (8 blocks of 5 trials/day) with a 5-minute rest between the blocks. Each block consisted of 5 trials, and each trial had a duration of 35 seconds. Participants in each group were allowed to first familiarize themselves with the task for 35 seconds. The participants in visual biofeedback group were required to make the diameter of a colored circle smaller by moving their COP. The diameter of the circle changed according to the distance between the real-time COP displacement and the moving target, growing as the COP displacement moved farther from the target and shrinking as the COP displacement moved closer to the target (Fig 3). Moreover, the color of the circle changed according to the position of the COP displacement to the target; a yellow color indicated the COP displacement shifted from the target in the forward direction (Fig 3A) and blue indicated the COP displacement shifted from the target in the backward direction (Fig 3B).

The participants in the auditory biofeedback group were required to modify the volume of a sound, reducing it as the distance between the COP displacement and the target decreased. In addition, the generated sound was higher-pitched (3000 Hz) as COP displacement shifted from the target in the forward direction (Fig 3A) and lower-pitched (1000 Hz) as COP displacement shifted from the target in the backward direction (Fig 3B). Both augmented sensory biofeedbacks were presented 75 milliseconds before and after the moving target reached the sine-wave inflection points (Fig 2) [18]. To inform the next direction of the moving target to participants, the visual biofeedback was displayed on the top of the monitor or the auditory biofeedback was sounded from the speaker in front of participants when the moving target reached the inflection point in a forward direction and vice versa. The perceptual magnitudes of visual biofeedback and auditory biofeedback were equalized according to Stevens' power law [25] as follows:

$$S = D^{\frac{1}{n}} \tag{1}$$

where $S$ is the perceptual magnitude, $D$ is the distance between the COP displacement and the target, and $n$ is defined by the sensory modality (visual: 0.9, auditory: 0.3). When the biofeedback was auditory, visual environmental cues were available and when the biofeedback was visual, auditory environmental cues were available.

## Outcome measures and statistical analysis

All signals were processed offline using MATLAB R2018b (The Mathworks Inc., Natick, MA, USA). Although the signals obtained in the test session had seven cycles, only six cycles were analyzed, excluding the first sine curve, in order to ignore the timing error due to the initiation of body sway. To evaluate the effects of motor learning, the mean and standard deviation (SD) of the distance between COP displacement and the target displacement were calculated for the 6 cycles in each trial. Then, the mean ($D_{mean}$) and SD ($D_{SD}$) across 5 trials in each block were calculated. Furthermore, the peak of COP displacement to both forward and backward direction in each cycle was normalized as a percentage for the stability limits toward both directions in each participant. Last, the difference between the peak COP displacement and the peak target displacement was calculated for the 6 cycles in each direction. The mean across 6 cycles in each direction was calculated, and then, the mean across 5 trials in each block was calculated as the value of "mean peak difference". This variable means a spatial error at the time intervals of biofeedback.

## (A) COP shifted from the target in the *forward* direction

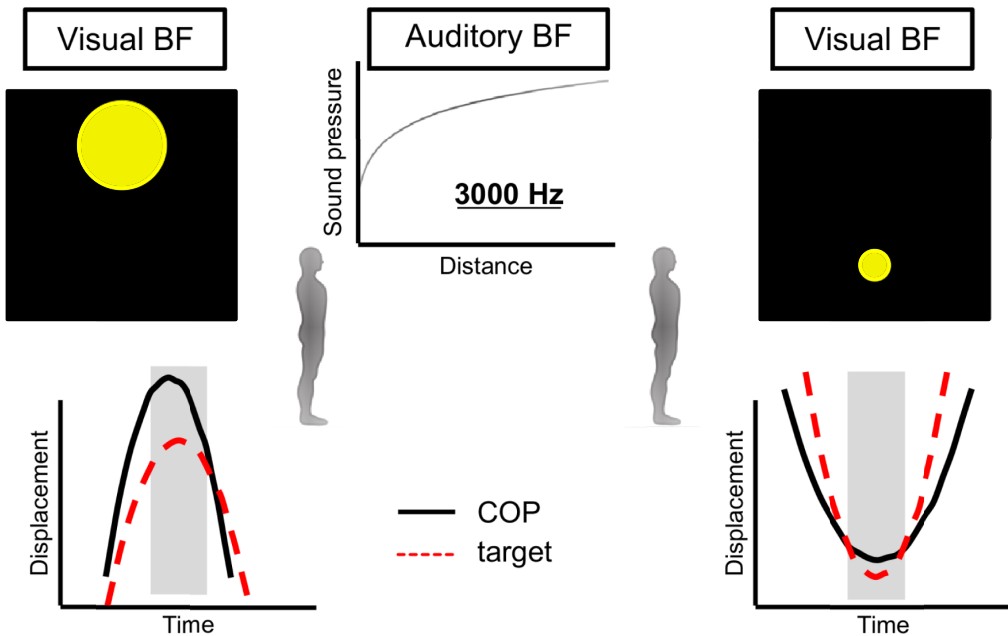

## (B) COP shifted from the target in the *backward* direction

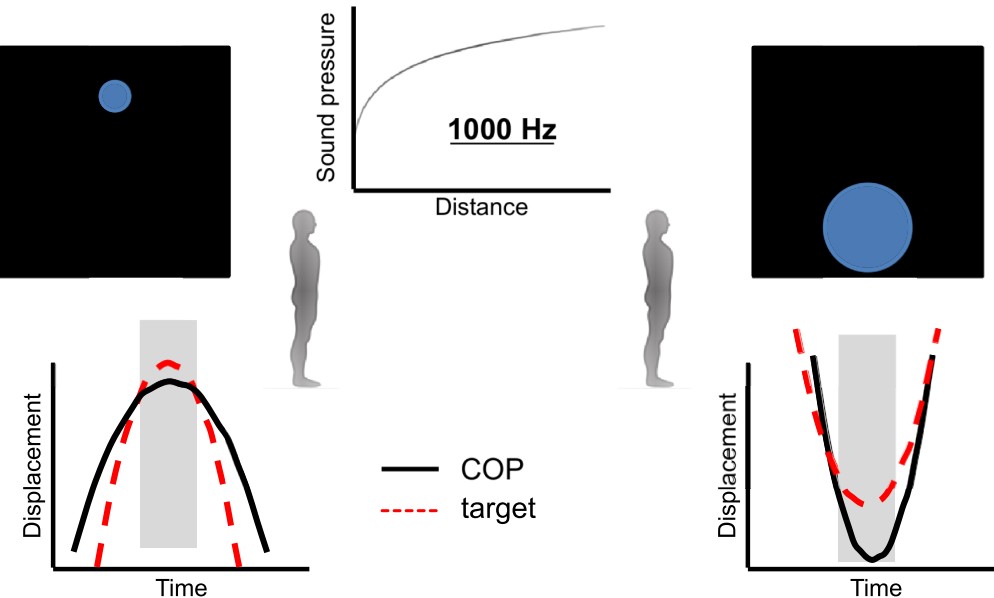

**Fig 3. Augmented sensory biofeedback.** For visual biofeedback, the diameter of the colored circle changed according to the distance between the real-time center of pressure (COP) displacement and the moving target in the anteroposterior direction. The larger yellow circle indicated that the COP displacement moved farther from the target and shifted from the target in the forward direction (A), while the larger blue circle indicated that the COP displacement moved farther from the target and shifted from the target in the backward direction (B). For auditory biofeedback, the volume changed according to the distance. The sound generated was higher-pitched (3000 Hz) as COP displacement shifted from the target in the forward direction (A) and lower-pitched (1000 Hz) as COP displacement shifted from the target in the backward direction (B). The visual biofeedback was displayed on the top of the monitor or the auditory biofeedback was sounded from the speaker in front of participants when the moving target reached the inflection point in a forward direction and vice versa. BF, augmented sensory biofeedback.

To evaluate the temporal domain of learning effects, coherence analysis was performed. Coherence is a function of the power spectral density of the COP displacement and the target signal, and the cross-power spectral density of the two signals. Magnitude-squared coherence is estimated as a function of sway frequency, with coherence values indicating the correspondence of the COP displacement signal to the target signal at each frequency bin ranging from 0, absence of any temporal relationship between the signals, to 1, perfect synchrony [26]. The spectral phase revealed the temporal relationship between two signals, expressed in degrees. The absolute synchronization between the two signals was represented by 0-degree phase lag, while the positive and negative values indicated that the COP displacement followed or preceded the target signal, respectively. To assess the temporal accuracy of postural control, we used the absolute value of phase lag [27]. The coherence function determined the magnitude-squared coherence estimate of the two signals using Welch's method with 6 segments of non-overlapping Hamming windows (frequency resolution = 0.01Hz) to average modified period grams. The peak coherence at 0.23 Hz was estimated on a subject-by-subject basis. The 95% confidence limit for the coherence spectrum was 0.45. The significant value was determined from the total segments per subject as follows:

$$1 - (0.05)^{1/(L-1)} \tag{2}$$

where $L$ is number of the total segments [28].

Two-way mixed-design ANOVA was used with the factors *Group* (visual biofeedback and auditory biofeedback) and *Test session* (pre-1, post-1, pre-2, post-2 and retention) to analyze possible differences in the above-mentioned parameters. Post-hoc analysis was performed using Bonferroni pairwise comparison. The relationships across the relative values of parameters were calculated using *Pearson's* correlation coefficient in each group. The relative values were calculated as the values on the retention test divided by those on the pre-1, and then were transformed to their natural logarithms to ensure the normal distribution. Thus, the relative values indicated the amount of learning effects. The statistical analysis for the outcome measure and correlation were processed using SPSS Statistics version 25.0 (IBM, Armonk, NY, USA). The statistical significance was set to $p < 0.05$.

## Results

No significant differences in participants' age, sex, height, weight, or foot length were found between the visual biofeedback and auditory biofeedback groups (Table 1).

### Spatial domain

A significant reduction of $D_{mean}$ and $D_{SD}$ was observed due to *Test session* ($D_{mean}$: $F_{4,\ 21} = 45.801$, $p < 0.001$; $D_{SD}$: $F_{4,\ 21} = 25.807$, $p < 0.001$; Table 2). In addition, these reductions were

**Table 1. The characteristics of the participants.**

|  | auditory BF (n = 11) | visual BF (n = 11) | *p*-value |
|---|---|---|---|
| Age (years) | 21.6±1.5 | 21.7±0.6 | 0.856 |
| Gender (male/female) | 7 / 4 | 6 / 5 | 0.361[a] |
| Height (cm) | 167.9±8.1 | 165.6±10.4 | 0.569 |
| Weight (kg) | 61.4±9.4 | 57.7±11.6 | 0.420 |
| Foot length (right) | 24.9±1.6 | 24.0±2.1 | 0.251 |

Groups compared using independent sample t-test or Chi-squared test and significance level of 0.05 (a: Chi-squared test). Mean ± Standard deviation; BF, augmented sensory biofeedback.

**Table 2. Results from two-way mixed-design ANOVA for each outcome measure in the spatial and temporal domain.**

| Outcomes | auditory BF, Mean (SD) | | | | | visual BF, Mean (SD) | | | | | Fixed factor | F value | p-value |
|---|---|---|---|---|---|---|---|---|---|---|---|---|---|
| | Pre-1 | Post-1 | Pre-2 | Post-2 | Retention | Pre-1 | Post-1 | Pre-2 | Post-2 | Retention | | | |
| **Spatial** | | | | | | | | | | | | | |
| $D_{mean}$ (mm) | 28.9 | 18.0 | 20.9 | 17.6 | 20.0 | 25.8 | 17.6 | 21.7 | 16.6 | 20.2 | **Test** | **45.801** | **< 0.001** |
| | (3.7) | (3.3) | (3.1) | (3.9) | (4.6) | (5.4) | (5.6) | (5.7) | (4.4) | (4.9) | Group | 0.185 | 0.671 |
| | | | | | | | | | | | Interaction | 1.538 | 0.199 |
| $D_{SD}$ (mm) | 29.1 | 18.8 | 22.0 | 18.7 | 21.8 | 26.9 | 18.2 | 23.8 | 18.5 | 21.5 | **Test** | **25.807** | **< 0.001** |
| | (4.5) | (3.7) | (4.2) | (3.0) | (5.8) | (6.5) | (7.6) | (7.8) | (7.7) | (6.6) | Group | 0.022 | 0.884 |
| | | | | | | | | | | | Interaction | 0.856 | 0.494 |
| Mean peak | 29.1 | 13.7 | 18.7 | 12.7 | 17.1 | 31.5 | 14.9 | 26.9 | 15.3 | 25.3 | **Test** | **52.563** | **< 0.001** |
| difference (%) | (7.1) | (5.9) | (7.2) | (2.2) | (5.6) | (8.2) | (3.5) | (6.8) | (2.7) | (6.0) | **Group** | **6.048** | **0.023** |
| | | | | | | | | | | | **Interaction** | **3.336** | **0.026** |
| **Temporal** | | | | | | | | | | | | | |
| Magnitude of | 0.958 | 0.973 | 0.973 | 0.980 | 0.980 | 0.961 | 0.966 | 0.962 | 0.961 | 0.969 | **Test** | **6.463** | **< 0.001** |
| coherence | (0.009) | (0.011) | (0.008) | (0.004) | (0.005) | (0.011) | (0.018) | (0.017) | (0.013) | (0.011) | **Group** | **9.676** | **0.006** |
| | | | | | | | | | | | **Interaction** | **3.254** | **0.016** |
| Phase lag | 16.1 | 11.6 | 11.3 | 9.2 | 8.4 | 17.2 | 16.0 | 17.3 | 15.3 | 19.9 | **Test** | **3.887** | **0.006** |
| (degrees) | (5.4) | (5.1) | (4.7) | (3.2) | (3.9) | (4.9) | (4.3) | (9.7) | (6.4) | (5.8) | **Group** | **9.249** | **0.006** |
| | | | | | | | | | | | **Interaction** | **5.554** | **0.002** |

Bold values indicate significant effects at $p < 0.05$.

BF, augmented sensory biofeedback; $D_{mean}$, the mean distance between the center of pressure (COP) and the moving target; $D_{SD}$, the standard deviation (SD) of the distance between the COP and the moving target.

similar across the two biofeedback groups ($D_{mean}$: $F_{1, 21} = 0.185$, $p = 0.671$; $D_{SD}$: $F_{1, 21} = 0.022$, $p = 0.884$; Table 2).

Post-hoc testing revealed that $D_{mean}$ and $D_{SD}$ in the retention trials were significantly decreased compared to pre-1 for both biofeedback groups ($D_{mean}$: auditory biofeedback, $p < 0.001$; visual biofeedback, $p = 0.001$; $D_{SD}$: auditory biofeedback, $p < 0.001$; visual biofeedback, $p = 0.019$; Fig 4A and 4B).

The mean peak difference between the COP and target displacements significantly decreased with *Test session* ($F_{4, 21} = 52.563$, $p < 0.001$), suggesting that the reduction of spatial error at peak occurred in both biofeedback groups (Table 2). A larger spatial error was observed in the visual biofeedback group compared to the auditory biofeedback group ($F_{1, 21} = 6.048$, $p = 0.023$; Table 2). A significant interaction between *Test session* and *Group* ($F_{4, 21} = 3.336$, $p = 0.026$; Table 2) was found for mean peak difference. Specifically, the post-hoc analysis showed that the decrease in mean peak difference was greater in the auditory biofeedback group compared to the visual biofeedback group at the pre-2 ($p = 0.012$), post-2 ($p = 0.020$), and retention ($p = 0.003$) (Fig 4C).

## Temporal domain

A significant interaction effect between *Test session* and *Group* was found on the magnitude of coherence and the phase lag ($F_{4, 21} = 3.254$, $p = 0.016$; Table 2). Specifically, the post-hoc analysis revealed that the auditory, but not the visual, biofeedback group showed a significant increase in the magnitude of coherence in the post-1 ($p = 0.001$), pre-2 ($p = 0.001$), post-2 ($p < 0.001$) and retention ($p < 0.001$) compared to baseline pre-1 (Fig 5A). The higher value of the magnitude of coherence indicated better success in tracking the target of the COP

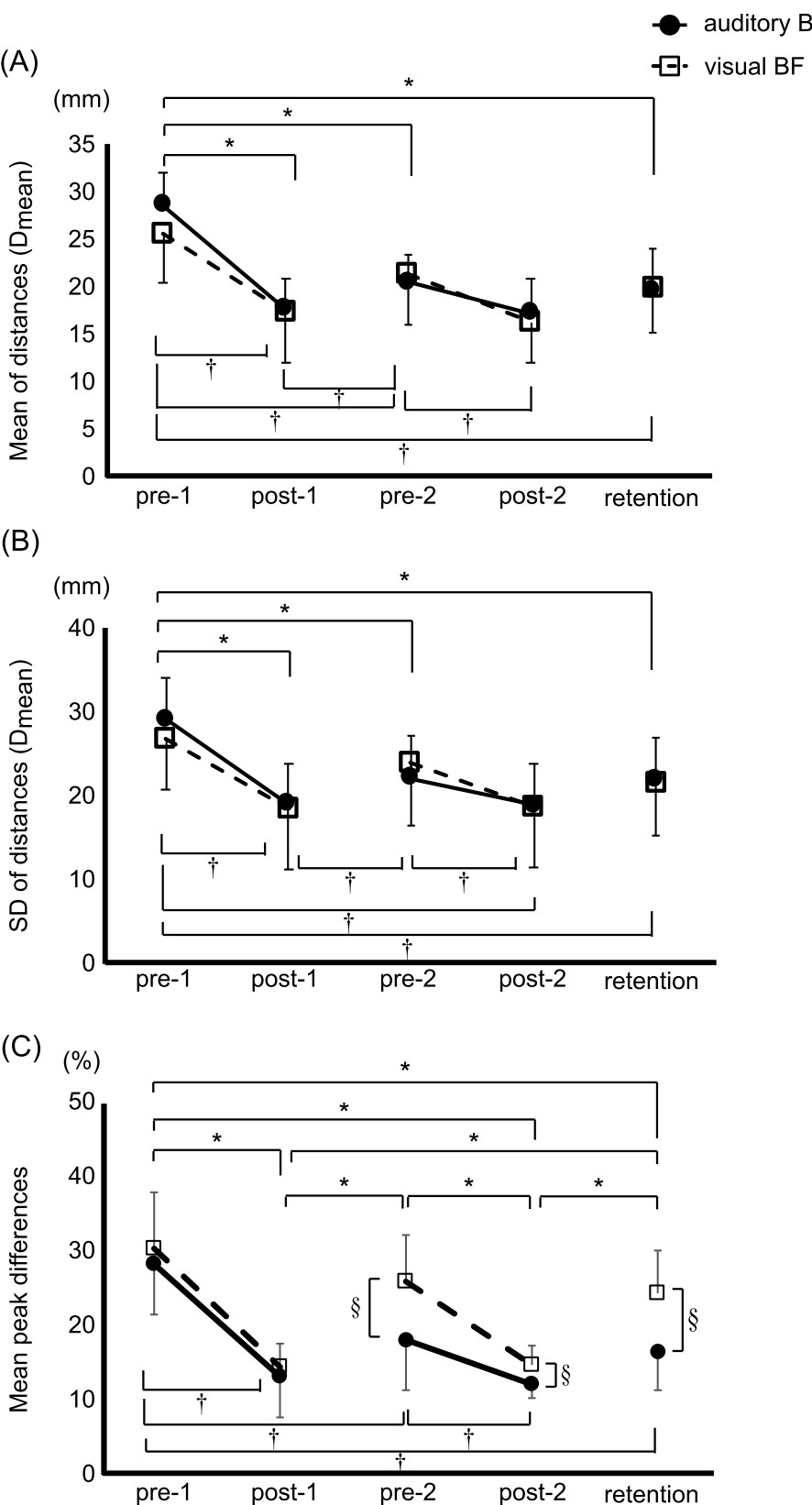

**Fig 4. Learning effects of both augmented sensory biofeedback training on outcomes in spatial domain.** Point plots of the mean (A) and standard deviation (SD) (B) of the distances between the center of pressure (COP) and the moving target, and the mean difference of peak movements between COP displacement and the moving target in the forward and backward directions (Mean peak difference) (C). The black circles represent the auditory augmented sensory biofeedback group, and the white squares represent the visual biofeedback group. Error bar shows a SD. * and † indicates a significant difference within auditory and visual biofeedback group, respectively ($p < 0.05$), and § indicates a significant difference between groups ($p < 0.05$). BF, augmented sensory biofeedback.

displacements. Moreover, the magnitude of coherence in the post-2 and retention in the auditory biofeedback group was significantly higher than that in the visual biofeedback group pre-1 in the auditory biofeedback group ($p < 0.001$ and $p = 0.017$, respectively). However, the visual biofeedback group showed no significant difference between pre-1 and the other test sessions (Fig 5A).

Further, a reduction of phase lag after auditory, but not visual, biofeedback training was found, revealed by a significant interaction between *Test session* and *Group* ($F_{4, 21} = 5.554$, $p = 0.001$; Table 2). Post-hoc analysis showed that the phase lag was significantly lower in the auditory biofeedback group than in the visual biofeedback group in the pre-1 ($p = 0.043$), post-2 ($p = 0.010$) and retention ($p < 0.001$) (Fig 5B). A smaller phase lag means a better temporal synchronization between the COP displacements and the target. In addition, the auditory biofeedback group showed significant reduction on the phase lag in the other test sessions post-1, pre-2, post-2, and retention compared to that in the pre-1 ($p = 0.011$, $p = 0.005$, $p < 0.001$, and $p < 0.001$, respectively). On the other hand, no significant difference was shown between pre-1 and the other test sessions in the visual biofeedback group (Fig 5B).

## Correlation

We found a significant relationship between the relative value of $D_{mean}$ and that of $D_{SD}$ in both biofeedback groups (auditory biofeedback: $r = 0.831$, $p = 0.002$; visual biofeedback: $r = 0.751$, $p = 0.008$). The relative value of the mean peak difference between COP and target displacement was significantly positively correlated with that of $D_{mean}$ in the auditory biofeedback group ($r = 0.606$, $p = 0.048$), but not in the visual biofeedback group ($r = 0.506$, $p = 0.112$) (Fig 6). No other significant relationships were found across the relative values.

## Discussion

Our findings reveal that discrete auditory biofeedback was more effective than discrete visual biofeedback for motor learning of voluntary postural sway (even after equalizing the perceptual magnitude of each type of biofeedback). The results of this study showed that both discrete biofeedback trainings improved postural control in the spatial domain under the no-feedback condition on the retention test compared to the pre-test (pre-1). However, only the discrete auditory biofeedback training enhanced postural control both in the temporal and spatial domains at the time intervals of biofeedback. Furthermore, the improvements in the spatial error at the time intervals of biofeedback significantly correlated with improvements in the spatial error over the whole trial in the auditory biofeedback group, but not in the visual biofeedback group.

As hypothesized, the learning effects of discrete auditory biofeedback training on postural control were superior to the discrete visual biofeedback training, particularly in the temporal domain and at the time intervals of biofeedback. One of the mechanisms explaining such differences between visual and auditory biofeedback may be a link between auditory and proprioceptive sensory systems. In fact, several studies demonstrated that auditory biofeedback enhanced multisensory integration and perceptual neural representation [2, 29–32]. For

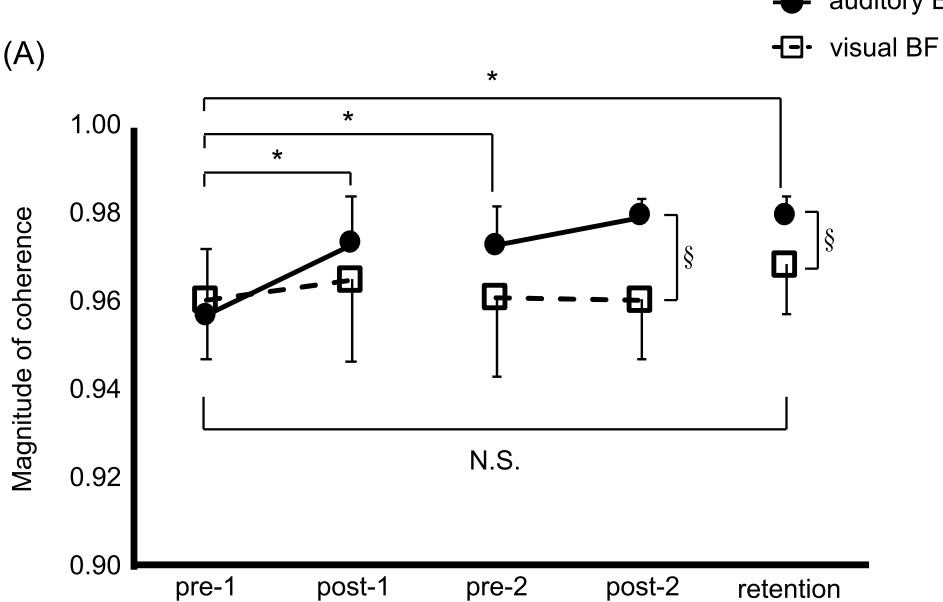

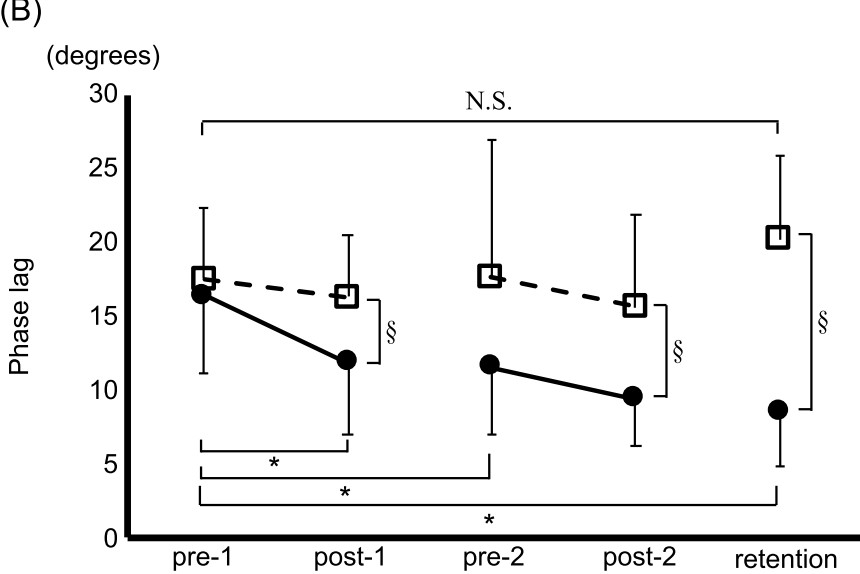

**Fig 5. Significant learning effects of auditory augmented sensory biofeedback training on outcomes in temporal domain.** Mean and standard deviation (SD) plots of two temporal measures: (A) magnitude of coherence and (B) phase lag. The black circles represent the auditory augmented sensory biofeedback group, and the white squares represent the visual biofeedback group. Error bar shows a standard deviation. *indicate a significant difference within auditory biofeedback group ($p < 0.05$) and § indicates a significant difference between groups ($p < 0.05$). N.S., non-significance; BF, augmented sensory biofeedback.

example, a study reported that auditory biofeedback training induced a significant enhancement of knee proprioception, shown as a lower knee repositioning error with auditory biofeedback [30,31]. In addition, the enhancement remained during the no-feedback condition immediately or 24-hour after the auditory biofeedback training [31]. These results suggested that, after auditory biofeedback, the participants not only learned to reproduce the movement precisely but also learned a more precise use of proprioceptive information from the knee

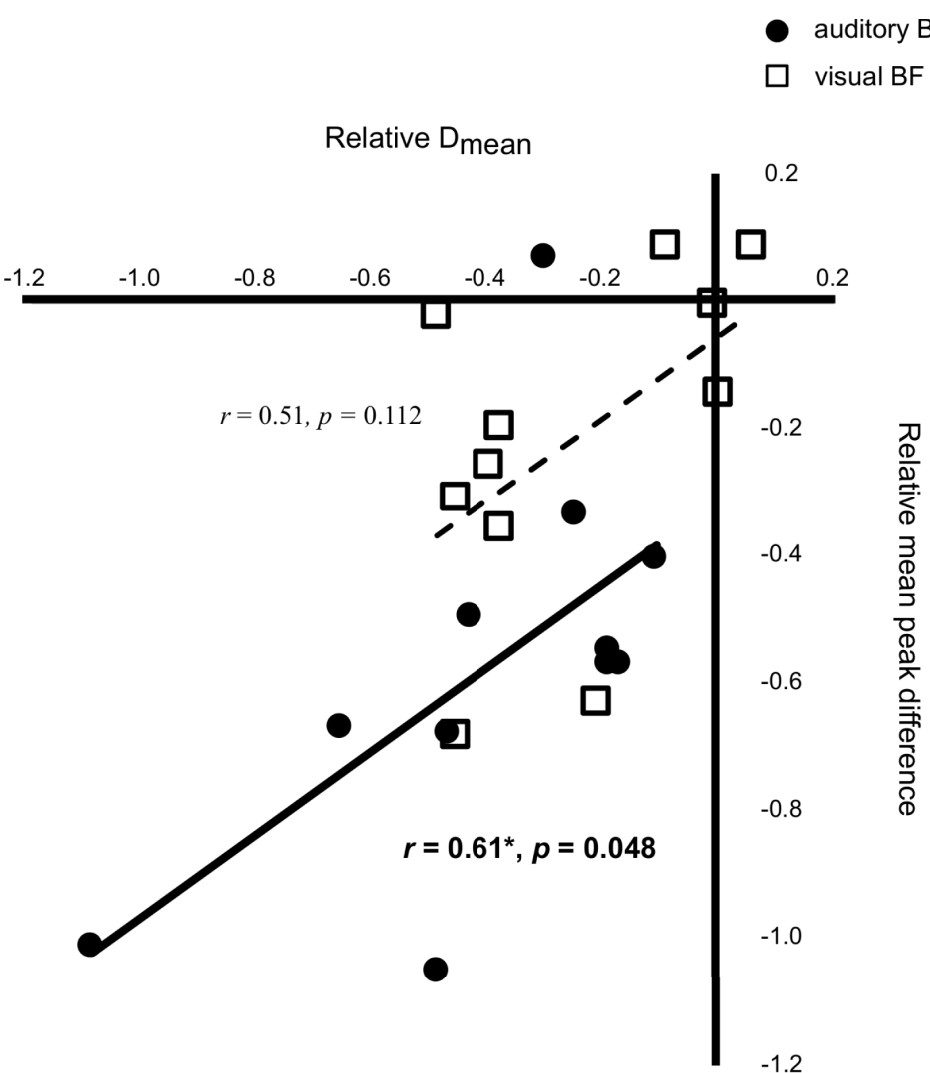

**Fig 6. Significant correlation between the improvements in both spatial errors in auditory augmented sensory biofeedback group.** Scatter plots of the relative value of the mean distance between the center of pressure (COP) and the moving target ($D_{mean}$) with the relative value of the mean difference of peak movements between COP displacement and the moving target in the forward and backward directions (Mean peak difference). Black circles represent the auditory augmented sensory biofeedback group, and white squares represent the visual augmented sensory biofeedback group. Transformed values to their natural logarithms are displayed, and *p*-value was calculated by a *Pearson's* correlation coefficient. BF, augmented sensory biofeedback.

joint. Likewise, some of the neuroimaging studies also supported the finding that auditory bio-feedback can promote coactivation in a broad network response to auditory and propriocep-tive information [19,32]. In contrast, visual biofeedback activates only the cortical areas playing a role in visuomotor transformation [19]. Therefore, one possibility why auditory bio-feedback was superior to visual biofeedback is that the auditory biofeedback system uses differ-ent learning strategies than the visual biofeedback system. In other words, visual biofeedback may promote a visuomotor transformation during augmented sensory biofeedback training, while auditory biofeedback may proceed motor learning by strengthening the intermodal cou-pling between auditory and proprioceptive information which contributes to the performance without augmented sensory biofeedback. Previous studies showed a stronger cognitive

involvement, represented by increased brain activation of prefrontal areas [19] and putamen [33], in performing a sensory-motor task when using auditory biofeedback compared to visual biofeedback. More cognitive involvement may enhance attention to intrinsic sensory information, especially proprioceptive information, and that may explain why the postural performance with auditory biofeedback was better than performance with visual biofeedback, even after the biofeedback was removed. Another explanation for the different learning strategies may be the different temporal accuracy between auditory and visual biofeedback. The stimulus-response reaction times for visual inputs are tens to one hundred milliseconds slower than that for auditory inputs [20,34]. Therefore, auditory biofeedback has an advantage in temporal resolution compared to visual biofeedback, which provides more temporal accuracy and reduced spatial error for auditory, than visual, biofeedback training. The slower visual processing results in delayed postural motor responses as apparent in the coherence values (lower values for the visual–moving the body less coherent with the stimulus) and in the phase (higher values for the visual–moving lagging behind the stimulus) (Fig 5).

We also found a significant reduction of spatial error under the no-feedback, retention, condition after either discrete visual or discrete auditory biofeedback training. In contrast, a previous study showed that continuous auditory biofeedback training, but not continuous visual biofeedback training, reduced spatial error for a voluntarily postural control task under the no-feedback condition even immediately after training [17]. This discrepancy could be explained by the type of biofeedback (continuous versus discrete). Consistent with our results, a recent study showed that discrete visual biofeedback training improved bimanual movements under the no-feedback condition after the biofeedback training similarly to discrete auditory biofeedback training, but not continuous visual biofeedback training [18]. Some researchers argue that reduced learning effects by visual biofeedback training are caused by "visual dominance" which is an excessive reliance on visual input with reduced other sensory contributions under the condition with visual biofeedback [35,36]. Therefore, reduced frequency of visual biofeedback during discrete biofeedback training, compared to continuous biofeedback training, may suppress the visual dominance, and then enhance spontaneous motor learning using proprioceptive input that contributes to the performance without biofeedback. This was supported by our results. In fact, the reduced mean peak difference was significantly associated with improvements of postural control in the spatial domain ($D_{mean}$) in the auditory biofeedback group only. $D_{mean}$ indicates the average spatial error for one trial, which consists of the area with and without augmented sensory biofeedback in the training session. Therefore, the significant correlation between reduced mean peak difference and improvements of postural control in the spatial domain result suggests that enhanced accuracy when using auditory biofeedback is responsible for the reduced spatial error under the no-feedback condition. On the other hand, no significant correlation between the improvements in mean peak difference and $D_{mean}$ was found in the discrete visual biofeedback training. This finding could suggest that reduced the whole spatial error in the discrete visual biofeedback group may be mainly caused by reduced spatial error of the area without augmented sensory biofeedback in the training session. In other words, the discrete visual biofeedback improves voluntary postural sway performance in the spatial domain mainly using spontaneous motor learning, not based on enhanced sensory information.

There are some limitations to this study. First, this experiment was performed with a small number of young participants. Therefore, we cannot be certain our findings would apply to people with neurologic disorders or older participants. Second, the learning effects by discrete biofeedback training were not directly compared with learning effects by the continuous biofeedback training. Last, neuroimaging should be investigated to understand the motor learning

mechanisms underlying the different learning effects of visual biofeedback versus auditory biofeedback training.

## Conclusions

This randomized trial demonstrated that discrete auditory biofeedback training was more effective than discrete visual biofeedback training for the motor learning of the voluntary postural sway task. Future studies should investigate the learning effects of the different types of visual and auditory biofeedback trainings in elderly persons or in people with sensory disorders. Furthermore, cortical activity and muscle synergies with sensory biofeedback training for postural control should be investigated in future studies.

## Acknowledgments

The authors thank our participants for generously donating their time to participate, and Norimasa Kakuya for helping with data collection and helping with study procedures.

## Author Contributions

**Conceptualization:** Naoya Hasegawa, Tadayoshi Asaka.

**Formal analysis:** Naoya Hasegawa.

**Funding acquisition:** Naoya Hasegawa, Fay B. Horak, Tadayoshi Asaka.

**Investigation:** Naoya Hasegawa.

**Methodology:** Naoya Hasegawa.

**Software:** Naoya Hasegawa, Kenta Takeda.

**Supervision:** Martina Mancini, Tadayoshi Asaka.

**Visualization:** Naoya Hasegawa.

**Writing – original draft:** Naoya Hasegawa.

**Writing – review & editing:** Kenta Takeda, Martina Mancini, Laurie A. King, Fay B. Horak, Tadayoshi Asaka.

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
