## [Decision Letter · Decision Letter 0]

17 Aug 2020

PONE-D-20-14779

Differential effects of visual versus auditory biofeedback training for postural control

PLOS ONE

Dear Dr. Asaka,

Thank you for submitting your manuscript to PLOS ONE. After careful consideration, we feel that it has merit but does not fully meet PLOS ONE’s publication criteria as it currently stands. Therefore, we invite you to submit a revised version of the manuscript that addresses the points raised during the review process.

Dear authors,

Despite one reviewer suggested acceptation, there are important concerns indicated by second reviewer. The authors need to improve introduction rationality and results presentation (improve the writing and sequence). Both aspects are important to consider the manuscript for publication. 

We look forward to receiving your revised manuscript.

Kind regards,

Fabio A. Barbieri, PhD

Academic Editor

PLOS ONE

Journal Requirements:

Reviewers' comments:

Reviewer's Responses to Questions

**Comments to the Author**

1. Is the manuscript technically sound, and do the data support the conclusions?

Reviewer #1: Yes

Reviewer #2: Partly

2. Has the statistical analysis been performed appropriately and rigorously? 

Reviewer #1: Yes

Reviewer #2: Yes

3. Have the authors made all data underlying the findings in their manuscript fully available?

Reviewer #1: No

Reviewer #2: Yes

4. Is the manuscript presented in an intelligible fashion and written in standard English?

Reviewer #1: Yes

Reviewer #2: Yes

5. Review Comments to the Author

Reviewer #1: The goal of this study was to determine if discrete auditory feedback during a body sway targeting task improved performance more than discrete visual feedback. The study was well designed, the methods were appropriate, and the manuscript is well written. The strengths of the paper include (1) random assignment of participants to two groups, and (2) the use of both spatial and temporal measures to assess performance. There are a few minor issues and one curiosity (#5) listed below. I would like to commend the authors for this work – it was a please to review this article. Overall, this manuscript is relevant and will be of interest to a wide range of researchers and clinicians.

Minor Issues:

1. Avoid acronyms, especially in the introduction/discussion – using the word feedback rather than BF doesn’t take much more space.

2. Line 373 – The results of the study are not simply ‘suggestive’, given the randomized trial, it was demonstrated that auditory feedback was more effective than visual feedback for the swaying task.

3. I recommend the authors reconsider the text on lines 315-317. Being the first to conduct an experiment is not relevant; adding knowledge to the field is relevant. If the authors choose to keep the wording about being first, I recommend they state they were the first to demonstrate that discrete auditory feedback was more effective at improving performance and learning (assuming no one else has demonstrated this).

4. Line 81 – extra ‘e’

5. This is a question I have, which may or may not be relevant for the text. Is there any indication that visual or auditory feedback have different cognitive demands? I would predict that, at least initially, auditory feedback would be more challenging due to the transformation of volume and pitch into sway magnitude and direction, whereas visual feedback does not (apparently) need to be transformed. Perhaps this distinction is related to improved learning – the auditory may be more attention-demanding, which may promote learning.

Reviewer #2: The present manuscript investigated the effectiveness of discrete visual versus auditory biofeedback (BF) to improve a postural tracking task. Twenty-two young participants were assigned to either a visual or auditory BF group. Participants were asked to shift their center of pressure (CoP) by voluntarily swaying forward and backward following a hidden, moving in a sinusoidal fashion and displayed intermittently. Results showed that, according to the authors, auditory BF was better than visual BF improving spatial and temporal relationship between CoP and target positioning. Based upon these results it was concluded that motor learning of postural control was improved by discrete auditory BF training.

Overall, the manuscript focuses on an important issue, which is related to improvement of postural positioning in a tracking task under different sensory cues displayed intermittently. The design and procedures seem to be sound, although a few and important issues still need to be clarified. Also the results and interpretation need to further explained. As a consequence, there a few issues that, as reviewer who is reading the manuscript for the first time, I would like to point out.

The first issue that needs to be revised and/or further clarified in the manuscript is the idea of “training for postural control”. Such usage does not reflect the essence of what is training. In my opinion, the training focuses on one particular behavior of tracking, visual or auditory, intermittent target. The movement involving the whole body is nominated, correctly, as postural control, but there is no measure or intention (at least from my view) of measuring postural control performance other than following the target. Using the broader scope such as training for postural control might furnish an equivocal idea that the training is towards improvement of postural control as a daily use task. Certainty, this is not the case and the manipulation and results cannot be used for such usage. Thus, there is the need to clarify and better refer, throughout the manuscript and including the title, to the task involved in the study rather than generalize as “postural control”.

Second, introduction needs to be improved in order to justify the rationality of the study. Why training effects would be potentially different between visual and auditory BF? Such clarification seems to be important even for preparing the reader for the proposed hypothesis. I am still wondering why was hypothesized that intermittent auditory BF would lead to learning but not visual BF. Yet, such learning effect would occur only in the temporal domain. Please further develop rationality for each of these issues even to support any discussion regarding the observed results.

Procedures seem to be sound, but there is the need to further describe the conditions and instructions for the participants. Regarding the conditions, when the stimulus was auditory, participants had visual cues available? When participants were performing the visual training, auditory cues from the environment were available. A detailed description regarding the available cues is critical in order to further understand and discuss possible differences between sensory cues.

Results are hard to follow, mainly for a couple of reasons. First, please refer in the text, because present the results where the reader can find the data. Refer to the Figure that the reader could see the results. Second, please clarify the statistical notation. It seems that statistical notation for auditory results are presented in the superior portion of the plot and for vision in the inferior part of it (I am not sure about this). For example, Figure 5B visual was not significant, but the note is nearby the symbol. On the other hand, auditory BF was significant and there is the notation indicating “NS”.

Finally, I do not agree with the interpretation and conclusion that only auditory BF improved postural tracking. In this case, there are a few aspects that must be clarified. My first concern is regarding the use and interpretation of the variables employed in the study. From my understanding, the most important variable indicating if the task was accomplished is the average and SD of distance (even mentioned – line 193-194). The mean peak difference is the error at the time interval of BF. Thus, data show that both sensory cues were used to improve the tracking, but more erratically in the visual condition. Such difference is even noticed in the coherence values (lower values for the visual – moving the body less coherent with the stimulus) and in the phase (higher values for the visual – moving lagging behind the stimulus). These are the different strategies that participants adopted in using different sensory cues. The question is why participants adopt these different strategies? Is the visual processing slower than the auditory? Is visual used more likely a confirmatory cue? In my point of view, the manuscript should discuss and clarify these issues, but not saying the visual training did not improve postural tracking because it did (at least from what I could get from Figure 4). Finally, I did not understand the usage of the correlation analysis. Yet, the number of participants and the obtained results do not allow for a clear cut interpretation of this possible relationship. Why does use it?

Minor issue:

- Please revise abstract, reducing its size and presenting results and conclusion properly

- Line 59-60: statement here contradicts the following sentence (lines 61-63). Please revise.

- Hypothesis must be justified.

- Please reference Figures in the text, before presenting results.

- Figure 1 is hard to follow with those boxes “WithBF” and “WithoutBF”.

- Please revise statistical notation in the Figures

6. PLOS authors have the option to publish the peer review history of their article (what does this mean?). If published, this will include your full peer review and any attached files.

Reviewer #1: No

Reviewer #2: No

---

## [Author Response · Author response to Decision Letter 0]

17 Sep 2020

Dear Reviewers and Editor,

We deeply thank the reviewers for their time and effort in peer-reviewing this manuscript. Your comments have been very useful and have helped to improve the manuscript. Below you can find a point-by-point response to each comment. All changes in the manuscript have been highlighted in yellow.

Reviewer #1

The goal of this study was to determine if discrete auditory feedback during a body sway targeting task improved performance more than discrete visual feedback. The study was well designed, the methods were appropriate, and the manuscript is well written. The strengths of the paper include (1) random assignment of participants to two groups, and (2) the use of both spatial and temporal measures to assess performance. There are a few minor issues and one curiosity (#5) listed below. I would like to commend the authors for this work – it was a please to review this article. Overall, this manuscript is relevant and will be of interest to a wide range of researchers and clinicians.

Minor Issues:

Point 1: Avoid acronyms, especially in the introduction/discussion – using the word feedback rather than BF doesn’t take much more space.

Response 1: We changed BF to biofeedback in the whole manuscript, except for figures and tables.

Point 2: Line 373 – The results of the study are not simply

‘suggestive’, given the randomized trial, it was demonstrated that auditory feedback was more effective than visual feedback for the swaying task.

Response 2: Thank you, we have modified the sentence in the conclusion.

“This randomized trial demonstrated that discrete auditory biofeedback training was more effective than discrete visual biofeedback training for the motor learning of the voluntary postural sway task.”(Page 24, Line 436)

Point 3: I recommend the authors reconsider the text on lines 315-317. Being the first to conduct an experiment is not relevant; adding knowledge to the field is relevant. If the authors choose to keep the wording about being first, I recommend they state they were the first to demonstrate that discrete auditory feedback was more effective at improving performance and learning (assuming no one else has demonstrated this).

Response 3: Thank you for the suggestion. The first paragraph of the discussion leads as follow: 

“Our findings reveal that discrete auditory biofeedback was more effective than discrete visual biofeedback for motor learning of voluntary postural sway (even after equalizing the perceptual magnitude of each type of biofeedback)” (Page 20, Line 352).

Point 4: Line 81 – extra ‘e’

Response 4: We apologize for the typo. The extra ‘e’ was removed.

Point 5: This is a question I have, which may or may not be relevant for the text. Is there any indication that visual or auditory feedback have different cognitive demands? I would predict that, at least initially, auditory feedback would be more challenging due to the transformation of volume and pitch into sway magnitude and direction, whereas visual feedback does not (apparently) need to be transformed. Perhaps this distinction is related to improved learning – the auditory may be more attention-demanding, which may promote learning.

Response 5: The reviewer brought up a good point. Previous studies have suggested that auditory feedback training requires more cognitive involvement than visual feedback training on upper limb task. However, no studies investigated on balance tasks. We have added the following sentences in the discussion: 

Page 21, Line 383: Previous studies showed a stronger cognitive involvement, represented by increased brain activation of prefrontal areas [29] and putamen [36], in performing a sensory-motor task when using auditory biofeedback compared to visual biofeedback. More cognitive involvement may enhance attention to intrinsic sensory information, especially proprioceptive information, and that may explain why the postural performance with auditory biofeedback was better than performance with visual biofeedback, even after the biofeedback was removed.

Reviewer #2

The present manuscript investigated the effectiveness of discrete visual versus auditory biofeedback (BF) to improve a postural tracking task. Twenty-two young participants were assigned to either a visual or auditory BF group. Participants were asked to shift their center of pressure (CoP) by voluntarily swaying forward and backward following a hidden, moving in a sinusoidal fashion and displayed intermittently. Results showed that, according to the authors, auditory BF was better than visual BF improving spatial and temporal relationship between CoP and target positioning. Based upon these results it was concluded that motor learning of postural control was improved by discrete auditory BF training.

Overall, the manuscript focuses on an important issue, which is related to improvement of postural positioning in a tracking task under different sensory cues displayed intermittently. The design and procedures seem to be sound, although a few and important issues still need to be clarified. Also the results and interpretation need to further explained. As a consequence, there a few issues that, as reviewer who is reading the manuscript for the first time, I would like to point out.

Point 1: The first issue that needs to be revised and/or further clarified in the manuscript is the idea of “training for postural control”. Such usage does not reflect the essence of what is training. In my opinion, the training focuses on one particular behavior of tracking, visual or auditory, intermittent target. The movement involving the whole body is nominated, correctly, as postural control, but there is no measure or intention (at least from my view) of measuring postural control performance other than following the target. Using the broader scope such as training for postural control might furnish an equivocal idea that the training is towards improvement of postural control as a daily use task. Certainty, this is not the case and the manipulation and results cannot be used for such usage. Thus, there is the need to clarify and better refer, throughout the manuscript and including the title, to the task involved in the study rather than generalize as “postural control”.

Response 1: We deeply appreciate the reviewer’s valuable comment. We changed “postural control” to “voluntary postural sway”.

Point 2: Second, introduction needs to be improved in order to justify the rationality of the study. Why training effects would be potentially different between visual and auditory BF? Such clarification seems to be important even for preparing the reader for the proposed hypothesis. I am still wondering why was hypothesized that intermittent auditory BF would lead to learning but not visual BF. Yet, such learning effect would occur only in the temporal domain. Please further develop rationality for each of these issues even to support any discussion regarding the observed results.

Response 2: Thank you for your important comments. We revised the last paragraph of the introduction.

Page 5, Line 87: The goal of this study was to investigate the learning effects of discrete auditory versus visual biofeedback to improve postural control, using a voluntary postural sway task [17]. A previous study using functional magnetic resonance imaging showed that brain activation increased in sensory-specific areas during visual biofeedback training. In contrast, brain activation gradually decreased over time with auditory biofeedback training [19]. These findings suggest that auditory biofeedback training may suppress reliance on augmented biofeedback during training unlike visual biofeedback training which requires sustained dependence on vision. Moreover, previous studies showed that auditory inputs are processed more quickly, shorter reaction times, compared to visual inputs for motor response [20-22]. Thus, auditory biofeedback would result in faster, more accurate influence on the temporal domain of postural control compared to visual biofeedback. Therefore, we hypothesized that discrete auditory biofeedback training would result in better learning effects than visual biofeedback, especially in the temporal domain for control of voluntary postural sway.

Point 3: Procedures seem to be sound, but there is the need to further describe the conditions and instructions for the participants. Regarding the conditions, when the stimulus was auditory, participants had visual cues available? When participants were performing the visual training, auditory cues from the environment were available. A detailed description regarding the available cues is critical in order to further understand and discuss possible differences between sensory cues.

Response 3: We apologize for the insufficient description for the conditions and instructions. We added the following to the methods. 

Page 11, Line 208: When the biofeedback was auditory, visual environmental cues were available and when the biofeedback was visual, auditory environmental cues were available.

Page 7, Line 128: After measuring the stability limits, participants were asked to perform the test and training sessions with the same stance and position of arms while maintaining attention on the monitor.

Point 4: Results are hard to follow, mainly for a couple of reasons. First, please refer in the text, because present the results where the reader can find the data. Refer to the Figure that the reader could see the results. Second, please clarify the statistical notation. It seems that statistical notation for auditory results are presented in the superior portion of the plot and for vision in the inferior part of it (I am not sure about this). For example, Figure 5B visual was not significant, but the note is nearby the symbol. On the other hand, auditory BF was significant and there is the notation indicating “NS”.

Response 4: We now describe each figure in the text. Also, we simplified Figure 4 and Figure 5 to be easier to understand. In addition, we added a table (Table 2) in the results section to explain the results from the two-way mixed-design ANOVA clearly.

Point 5: Finally, I do not agree with the interpretation and conclusion that only auditory BF improved postural tracking. In this case, there are a few aspects that must be clarified. My first concern is regarding the use and interpretation of the variables employed in the study. From my understanding, the most important variable indicating if the task was accomplished is the average and SD of distance (even mentioned – line 193-194). The mean peak difference is the error at the time interval of BF. Thus, data show that both sensory cues were used to improve the tracking, but more erratically in the visual condition. Such difference is even noticed in the coherence values (lower values for the visual – moving the body less coherent with the stimulus) and in the phase (higher values for the visual – moving lagging behind the stimulus). These are the different strategies that participants adopted in using different sensory cues. The question is why participants adopt these different strategies? Is the visual processing slower than the auditory? Is visual used more likely a confirmatory cue? In my point of view, the manuscript should discuss and clarify these issues, but not saying the visual training did not improve postural tracking because it did (at least from what I could get from Figure 4). 

Response 5: Thank you for your thoughtful comments. We don’t seem that the visual biofeedback was more confirmative compared to auditory biofeedback in this study. This is because we tried to equalize the perceptual magnitude of both visual and auditory biofeedback using Stevens’ power law (Page 11, Line 203). However, previous studies showed the different cognitive challenges and temporal resolutions between visual and auditory biofeedback, which could lead to different strategies. 

As stated above, we added the impact of different cognitive challenges and temporal resolutions between auditory and visual biofeedback as follows: 

Page 21, Line 383: Previous studies showed a stronger cognitive involvement, represented by increased brain activation of prefrontal areas [19] and putamen [33], in performing a sensory-motor task when using auditory biofeedback compared to visual biofeedback. More cognitive involvement may enhance attention to intrinsic sensory information, especially proprioceptive information, and that may explain why the postural performance with auditory biofeedback was better than performance with visual biofeedback, even after the biofeedback was removed. Another explanation for the different learning strategies may be the different temporal accuracy between auditory and visual biofeedback. The stimulus-response reaction times for visual inputs are tens to one hundred milliseconds slower than that for auditory inputs [20,34]. Therefore, auditory biofeedback has an advantage in temporal resolution compared to visual biofeedback, which provides more temporal accuracy and reduced spatial error for auditory, than visual, biofeedback training. The slower visual processing results in delayed postural motor responses as apparent in the coherence values (lower values for the visual – moving the body less coherent with the stimulus) and in the phase (higher values for the visual – moving lagging behind the stimulus) (Fig .5).

Point 6: Finally, I did not understand the usage of the correlation analysis. Yet, the number of participants and the obtained results do not allow for a clear cut interpretation of this possible relationship. Why does use it?

Response 6: We tried to demonstrate that the visual biofeedback improved the performance using spontaneous learning effects, not via visual biofeedback. In fact, results suggest that the improved spatial error was mainly reached by the improved spatial error of the area without biofeedback in the training session (white area in Fig. 2). 

As stated above, we revised the sentences in the discussion as follows: 

Page 23, Line 413: In fact, the reduced mean peak difference was significantly associated with improvements of postural control in the spatial domain (Dmean) in the auditory biofeedback group only. Dmean indicates the average spatial error for one trial, which consists of the area with and without augmented sensory biofeedback in the training session. Therefore, the significant correlation between reduced mean peak difference and improvements of postural control in the spatial domain result suggests that enhanced accuracy when using auditory biofeedback is responsible for the reduced spatial error under the no-feedback condition. On the other hand, no significant correlation between the improvements in mean peak difference and Dmean was found in the discrete visual biofeedback training. This finding suggests that reduced the whole spatial error in the discrete visual biofeedback group may be mainly caused by reduced spatial error of the area without augmented sensory biofeedback in the training session. In other words, the discrete visual biofeedback improves voluntary postural sway performance in the spatial domain mainly using spontaneous motor learning, not based on enhanced sensory information.

Minor issue:

Point 7: Please revise abstract, reducing its size and presenting results and conclusion properly

Response 7: Thank you. We revised the abstract as your suggestion.

Point 8: Line 59-60: statement here contradicts the following sentence (lines 61-63). Please revise.

Response 8: We apologize for the contradiction. We re-wrote the phrase as follows: 

Page 4, Line 63: Although a few studies have reported the effects of visual or auditory biofeedback training on postural control, in our knowledge, only our previous study reported that one modality was better than the other by direct comparison.

Point 9: Hypothesis must be justified.

Response 9: We revised the last paragraph of the introduction as Response 2. 

Point 10: Please reference Figures in the text, before presenting results.

Response 10: We referred each figure in the text. 

Point 11: Figure 1 is hard to follow with those boxes “WithBF” and “WithoutBF”.

Response 11: We modified Figure 1. 

Point 12: Please revise statistical notation in the Figures

Response 12: Thank you, we modified Figure 4 and Figure 5 as Response 4.

---

## [Decision Letter · Decision Letter 1]

14 Dec 2020

Differential effects of visual versus auditory biofeedback training for voluntary postural sway

PONE-D-20-14779R1

Dear Dr. Asaka,

We’re pleased to inform you that your manuscript has been judged scientifically suitable for publication and will be formally accepted for publication once it meets all outstanding technical requirements.

Kind regards,

Fabio A. Barbieri, PhD

Academic Editor

PLOS ONE

Additional Editor Comments (optional):

Reviewers' comments:

Reviewer's Responses to Questions

**Comments to the Author**

1. If the authors have adequately addressed your comments raised in a previous round of review and you feel that this manuscript is now acceptable for publication, you may indicate that here to bypass the “Comments to the Author” section, enter your conflict of interest statement in the “Confidential to Editor” section, and submit your "Accept" recommendation.

Reviewer #2: All comments have been addressed

2. Is the manuscript technically sound, and do the data support the conclusions?

Reviewer #2: Yes

3. Has the statistical analysis been performed appropriately and rigorously? 

Reviewer #2: Yes

4. Have the authors made all data underlying the findings in their manuscript fully available?

Reviewer #2: Yes

5. Is the manuscript presented in an intelligible fashion and written in standard English?

Reviewer #2: Yes

6. Review Comments to the Author

Reviewer #2: The revised version of the manuscript has been improved with all the suggested changes made. The authors also, based upon the suggestions from the reviewers, made substantial changes in the presentation and, most importantly, in the interpretation of the results, contributing significantly to the knowledge in the field. Thus, I congratulate with the authors for the submission and the presented manuscript.

7. PLOS authors have the option to publish the peer review history of their article (what does this mean?). If published, this will include your full peer review and any attached files.

Reviewer #2: **Yes: **Jose A. Barela

---

## [Editor Report · Acceptance letter]

16 Dec 2020

PONE-D-20-14779R1 

Differential effects of visual versus auditory biofeedback training
for voluntary postural sway 

Dear Dr. Asaka:

I'm pleased to inform you that your manuscript has been deemed suitable for publication in PLOS ONE. Congratulations! Your manuscript is now with our production department. 

Kind regards, 

on behalf of

Dr. Fabio A. Barbieri 

Academic Editor

PLOS ONE